# Occurrence of late-apoptotic symptoms in porcine preimplantation embryos upon exposure of oocytes to perfluoroalkyl substances (PFASs) under *in vitro* meiotic maturation

**Anna Leclercq**[1]*, **Petter Ranefall**[2], **Ylva Cecilia Björnsdotter Sjunnesson**[1], **Ida Hallberg**[1]

**1** Division of Reproduction, Department of Clinical Sciences & the Centre for Reproductive biology in Uppsala, Swedish University of Agricultural Sciences, Uppsala, Sweden, **2** Department of Information Technology, and SciLifeLab BioImage Informatics Facility, Uppsala University, Uppsala, Sweden

* anna.leclercq@slu.se

## Abstract

The objectives of this study were to evaluate the effect of perfluoroalkyl substances on early embryonic development and apoptosis in blastocysts using a porcine *in vitro* model. Porcine oocytes (N = 855) collected from abattoir ovaries were subjected to perfluorooctane sulfonic acid (PFOS) (0.1 μg/ml) and perfluorohexane sulfonic acid (PFHxS) (40 μg/ml) during *in vitro* maturation (IVM) for 45 h. The gametes were then fertilized and cultured *in vitro*, and developmental parameters were recorded. After 6 days of culture, resulting blastocysts (N = 146) were stained using a terminal deoxynucleotidyl transferase dUTP nick end labeling (TUNEL) assay and imaged as stacks using confocal laser scanning microscopy. Proportion of apoptotic cells as well as total numbers of nuclei in each blastocyst were analyzed using objective image analysis. The experiment was run in 9 replicates, always with a control present. Effects on developmental parameters were analyzed using logistic regression, and effects on apoptosis and total numbers of nuclei were analyzed using linear regression. Higher cell count was associated with lower proportion of apoptotic cells, *i.e.*, larger blastocysts contained less apoptotic cells. Upon PFAS exposure during IVM, PFHxS tended to result in higher blastocyst rates on day 5 post fertilization (p = 0.07) and on day 6 post fertilization (p = 0.05) as well as in higher apoptosis rates in blastocysts (p = 0.06). PFHxS resulted in higher total cell counts in blastocysts (p = 0.002). No effects attributable to the concentration of PFOS used here was seen. These findings add to the evidence that some perfluoroalkyl substances may affect female reproduction. More studies are needed to better understand potential implications for continued development as well as for human health.

**Data Availability Statement:** All relevant data are within the paper and its Supporting Information file.

**Funding:** Funding was awarded to the following authors: YS- The Swedish Research Council for Environment, Agricultural Sciences and Spatial Planning, url: https://formas.se/ (FORMAS grant no. 942-2015-476). IH- Stiftelsen Nils Lagerlöfs fund, url: https://www.ksla.se/ (KSLA grant no. GFS2021-0031) PR- The BioImage Informatics Facility is funded by SciLifeLab, National Microscopy Infrastructure, url: https://www.vr.se/om-vetenskapsradet/organisation/amnesrad-rad-och-kommitteer/radet-for-forskningens-infrastrukturer.html (grant no VR-RFI 2019-00217), and the Chan-Zuckerberg Initiative AL, YS- The Cells for Life Platform and Developmental Biology Platform at the Swedish University of Agricultural Sciences provided the IVF facilities. A scholarship was also provided by the VH faculty at the Swedish University of Agricultural Sciences (url: https://www.slu.se/en/). The funders had no role in study design, data collection and analysis, decision to publish, or preparation of the manuscript.

**Competing interests:** The authors have declared that no competing interests exist.

## Introduction

Infertility, defined as the inability to conceive within 12 months of actively trying, is an issue estimated to affect about ten percent of women aged 20–44 worldwide [1]. Even after medical investigation, the reason for infertility remains unknown in approximately 25–30% of cases [2, 3]. In addition to factors concerning genetics and lifestyle, the potential effects of endocrine disrupting chemicals (EDCs) on fertility has gained attention in later years [4]. Per- and poly-fluoroalkyl substances (PFASs) is one group of chemicals which is of particular concern due to their widespread distribution and persistence in the environment [5]. PFASs constitute a large group of chemicals with surfactant properties that are present in commonly used products such as fire-fighting foam, textiles and packaging materials [6]. Humans are mainly subjected to PFASs via contaminated foodstuff. Inhalation of particles and direct contact are also known routes of exposure [7–9]. PFASs can be detected in blood of the general population worldwide [10–12], as well as in foetuses [13] and in ovarian follicular fluid [14]. Blood concentrations of PFASs in residents of heavily contaminated areas, and individuals subjected to occupational exposure, have been found to be substantially higher than in the general population [15, 16].

Exposure to PFASs has been associated with negative effects on health, including increased serum cholesterol levels [17, 18], decreased response to vaccines [19], and lower birth weights in children [20]. In the human general population as well as in heavily exposed groups.

Even though manufactured and used for over 50 years, potential effects of PFASs on female reproduction are incompletely understood [21, 22]. Human cohorts have shown an association between exposure and early onset of menopause [23], alterations of the menstrual cycle [24], prolonged time to pregnancy [25] and sporadic first trimester miscarriages [20].

The process of forming a healthy offspring starts with germ cells, where the female contributes with the oocyte. The quality, or developmental competence, of oocytes affect survival rates of early embryos as well as the establishment of pregnancy and subsequent development [26]. Oocyte quality is determined during the complex process of folliculogenesis. The final stages of oocyte maturation constitute a particularly sensitive period in time where critical events take place.

There is only limited information regarding potential oocyte toxicity of PFASs. In animal models, oocyte toxicity has been observed in the mouse [27] and pig [28]. It has also previously been shown that development of blastocysts is affected upon exposure during oocyte maturation in the bovine using an *in vitro* model [29].

In this study, we aimed to evaluate the effects of exposure to perfluorooctane sulfonic acid (PFOS) and perfluorohexane sulfonic acid (PFHxS) during the final stages of oocyte maturation and the consequences for the developmental competence regarding blastocyst formation using a porcine *in vitro* model.

Previously, PFOS has been associated with increased rates of apoptosis in zebrafish and xenopus embryos [30–32]. Apoptosis is commonly described as programmed cell death and can, for instance, occur as a response to toxicity [33]. In the bovine blastocyst, PFASs appear to alter genes associated with apoptotic pathways [29]. Therefore, we also wished to assess the effects of PFASs on cell count and proportion of apoptotic cells in resulting blastocysts, by using a terminal deoxynucleotidyl transferase dUTP nick end labeling (TUNEL) staining assay and objective image analysis.

## Materials and method

### Project overview

For this project, porcine oocytes obtained from abattoir ovaries were used. A total of 855 oocytes were run in nine replicates of *in vitro* embryo production according to standard

procedures. Selected oocytes were divided into three equally large groups; PFOS group, PFHxS group, and control group (range: 28–40 oocytes/group). Oocytes were matured with the addition of PFOS (0.1 µg/mL) and PFHxS (40 µg/mL) respectively, for a total of 45 h. They were then fertilized using frozen thawed semen, where the same boar was used throughout the experiment. During the culture process, cleavage rate and cleavage rate above 2 cells were recorded 48 h post fertilization (hpf).

Number of blastocysts, as well as developmental stage and grade, were documented on day 5 and 6 post fertilization (pf), respectively. On day 6 pf, blastocysts were collected to be fixated and stained using 4′,6-diamidino-2-phenylindole (DAPI) and terminal deoxynucleotidyl transferase dUTP nick end labeling (TUNEL) stains to evaluate proportion apoptotic cells and cell count in blastocysts.

## Media and reagents

Porcine oocyte maturation medium (POM), porcine fertilization medium (PFM), and porcine zygote medium (PZM) were purchased from the Research Institute for the Functional Peptides, FHK Fujihura Industry Co Ltd, Osaka, Japan. Wash media and wash media with heparin were produced on site (wash media: gentamycine sulphate 10 µg/ml, L-glutamine MW 146.14 1 mM, PVA 3 µg/ml in Hepes TCM 199; wash media with heparin: heparin 20 U/ml in basic wash media). Commercial media were always pre-equilibrated for at least 2 hours in 38˚C and 5.5% $CO_2$ before use. For all *in vitro* production (IVP) procedures, four-well nunclon plates were used.

## Exposure

For exposure, PFOS (potassium salt >98%, CAS 1763-23-1) and PFHxS (potassium salt >98%, CAS 355-46-4) dissolved in molecular grade water was added to the maturation media of the respective exposed groups to reach the concentration of 0.1 µg/mL (PFOS) or 40 µg/mL (PFHxS). These concentrations were chosen based on results from previous experiments [29, 34]. In the control, vehicle (molecular grade water) was added in the same volume as the stock solutions containing PFOS/PFHxS. Stock solutions were stored at 4˚C protected from light during the course of the experiment. Due to persistence of the compounds used, significant degradation was not expected during storage. The stock solution concentrations of PFOS and PFHxS were validated using mass-spectrometry which is described elsewhere [29, 34].

## Collection of ovaries and maturation of oocytes

Ovaries from gilts intended for human consumption were collected at an abattoir in Uppsala, Sweden and transported (range, transportation time: 1.5–3 h) in 0.9% saline at 31˚C (range 30–32˚C) to the IVF laboratory. Ovaries were subsequently rinsed with 35˚C NaCl (0.9%), and poured into fresh NaCl (0.9%). Follicular fluid was aspirated from follicles measuring 3–8 mm in diameter using 5 mL syringes and 20 gauge cannulas, and transferred to approximately 7–10 mL of wash media with heparin (kept at 38˚C) in 25 mL tubes. Aspirated oocytes were allowed to settle to the bottom of the 25 mL tubes, and were then transferred to 60 mm petri dishes and covered with wash medium. Using stereo microscopes, oocytes with an even cytoplasm and at least two cumulus cell layers were chosen [35] and washed through three 30 mm petri dishes with wash media. Selected oocytes were randomly allocated into 3 morphologically and numerically equal groups (*i.e.* PFOS group, PFHxS group, and control group), which were kept separated and treated equally (with the exception of exposures) for the remainder of IVP including *in vitro* maturation, fixation and staining processes. For technical reasons, treatments were not blinded. Oocyte groups were washed in 460 µL of POM medium each, and

subsequently, transferred in 20 μL to POM medium with the addition of PFOS (0.1 μg/mL) and PFHxS (40 μg/mL), respectively. For details, see previous section. The corresponding amount of sterile water (used as vehicle) was added to the control group. The POM medium used in this step was, additionally, enriched with FSH (follicle stimulating hormone, 0.05 IU/mL) (FSH Porcine, OOPA00171, Insight Biotechnology, Middlesex, United Kingdom), LH (luteinizing hormone, C = 0.05 IU/mL) (LH Protein, OOPA00173, Insight Biotechnology), and dibutyryladenosine cyclic monophosphate (dbcAMP, 1mM) (dibutyryl-cAMP, sodium salt, 1698950, Biogems, Westlake Village, United States). Oocyte groups were matured in 5.5% $CO_2$ and 38.5˚C for 22 h. They were then transferred in 20 μL to 480 μL each of fresh POM medium with PFOS (0.1 μg/mL), PFHxS (40 μg/mL), and sterile water (respectively) and further matured (now without LH, FSH or dbcAMP) in 5.5% $CO_2$ and 38.5˚C for 23 hours.

## Fertilization in vitro

For all fertilizations, frozen thawed semen from the same boar, stored in in plastic 0.5 mL straws, was used. Prior to each fertilization, one straw was thawed for 30 seconds in 35˚C tap water. Motility of spermatozoa was controlled using a stereo microscope by putting 1 drop of thawed sperm directly on a glass slide. The semen was then poured into, and mixed with, 4 mL of PFM. Two mL of the sperm dilution was placed on top of 4 mL of room temperature single layer colloid (SLC) [36] in a centrifuge tube. The colloid and sperm were then centrifuged for 20 minutes at $300 \times g$. Supernatant sperm and excess fluid was disposed of, so that only a sperm pellet was left on the bottom of the centrifuge tube. The pellet was transferred to 0.75 mL of PFM. Sperm was counted in room temperature using a light microscope and a Bürcher chamber. Oocyte groups (now matured for 45 h) were washed in 480 μL/each of PFM, and subsequently transferred (in 20 μL/group) to 400 μL/each of PFM, to reach a final volume of 500 μL. Sperm dilution with adjusted concentration ($1.2 \times 10^6$) was added to each oocyte group. The oocytes and spermatozoa were incubated together in 5.5% $CO_2$ and 38.5˚C for 24 h.

## Culture and assessment of embryo development

Presumed zygotes were denuded by gentle pipetting in 4-wells containing 500 μL of wash media/well. They were then washed in 480 μL/each of PZM, and subsequently transferred (in 20 μL/group) to 480 μL of PZM, with a top layer of 400 μL of IVF oil (IVF Bioscience, Falmouth, United Kingdom), each. The zygotes were incubated in 38.5˚C, 5.5% $CO_2$ and 6% $O_2$ for 6 days. During the culture process, embryo development parameters were assessed (see below).

Forty-eight hours after fertilization, percentage of cleaved zygotes (all) and percentage of cleaved zygotes (above 2 cells) were documented for each group. On day 5 and 6 pf (respectively) number of blastocysts, as well as stage and grade of every individual blastocyst, were documented. Stages assigned were as follows: early blastocyst, blastocyst, expanding blastocyst, and hatching blastocyst [37]. Grade scores assigned were as follows: grade 1; excellent or good quality, grade 2; fair quality, grade 3; poor quality, grade 4; dead or degenerating [37]. Half grades were used when appropriate (*e.g.* grade 1.5).

## Fixation and staining

On day 6 pf, blastocysts were fixated overnight in 2% paraformaldehyde at 4˚C. After fixation, they were washed $\times$ 2 in phosphate buffered saline (PBS) with 0.1% polyvinyl alcohol (PVA).

A TUNEL staining kit (*In Situ* Cell Detection kit, TMR red, 12156792910, Roche, Mannheim, Germany) was used to stain apoptotic nuclei. To stain all nuclei, anti-fade mounting

medium with DAPI (Vectashield with DAPI, H-1200, Vector Laboratories, Burlingame, United States) was used. Staining procedures were executed according to the manufacturer´s instructions. The groups of stained blastocysts were mounted onto black-well plates (Diagnostic microscope slides 6.7 mm, ER-208B-CE24, Thermo Scientific, Braunschweig, Germany) in 2 μL PBS with 0.1% PVA and 2 μL anti fade mounting medium with DAPI. Slides were sealed using cover glasses and nail polish. Sealed slides were stored in darkness and 4˚C until confocal laser scanning procedures (see below).

For each staining session (n = 4), 2 of the blastocysts (chosen randomly) were sacrificed for use as positive and negative controls for the TUNEL assay. The control blastocysts were incubated in deoxyribonuclease (DNase I, 1073395, Qiagen, Hilden, Germany) solution (0.1 UI/μL in tris buffer) for 1 h at 37˚C in darkness, according to instructions. With the exception of the negative control not being subjected to the TUNEL reaction, controls were treated under the same circumstances as the remaining blastocysts.

## Confocal microscopy

All stained blastocysts were imaged as z-stacks with the use of a confocal microscope (LSM 800, Zeiss, Oberkochen, Germany). Section thickness was set to 2 μm. The 20x objective as well as lasers 405 and 561 were used for all blastocysts. Snap images were captured from the positive and negative controls in both channels. Treatments were blinded during confocal laser scanning procedures and image analysis.

## Image analysis

A total of146 embryos were included in the image analysis. An apoptotic nucleus was defined as: a nucleus within the blastocyst area, stained with both DAPI and TUNEL stains. TUNEL-positive cells were segmented and identified using a macro developed for ImageJ [38] as previously described [29]. In brief, the basis for the apoptosis segmentation was an iterative 3D version of the Per Object Ellipsefit (POE) algorithm. The original POE method computes local threshold levels for each object, where the threshold level is set to optimise the ellipse fit of the object, given that it fulfils input criteria of minimum and maximum diameters, also available in a 3D version [39]. To avoid complications with threshold settings when darker objects touches brighter objects, we developed an iterative version of the 3D POE method [29]. In brief, this version was set up as follows: first, the resulting mask from the 3D POE was kept as a "seed" image. This contained the segmented objects but eroded two voxels. Secondly, the segmented objects were masked from the raw image by making the corresponding voxels black. Subsequently, 3D POE was used to segment objects in the masked image. The resulting mask was merged with the previously segmented objects and the "seed" image was merged with the previous "seed" image. These steps were repeated until no more objects were segmented from the masked image. The objects in the resulting mask were separated using the 3D Watershed Split function [40] in ImageJ, with the combined mask as input, and the combined "seeds" as seeds. The nuclei were segmented by applying a global grey-level thresholding based on object size, SizeIntervalPrecision (SIP) [41], followed by watershed separation using the 3D Watershed Split function [40] in ImageJ, with local maxima in the 3D Distance Map of the initial threshold as seeds.

Manual validation of the automated image analysis was performed for all blastocysts by comparing the TUNEL z-stacks to the corresponding macro-derived images, and dead or degenerated embryos (n = 6) were excluded from the analysis, resulting in 140 embryos being included in further analyses.

## Statistical analysis

Statistical analyses were conducted in RStudio for R (R i386, 4.0.5). The effect of treatment (PFOS, PFHxS) compared to control on developmental parameters (proportions of cleaved, cleaved beyond 2 cell stage and proportion of blastocysts at day 5 and 6) was calculated using mixed effect logistic regression with binary distribution (glmer model of the lme4 package). Groups were weighted for size and replicate was included as a random effect. Day 5 and 6 blastocysts were considered repeated measurements and hence only one model for blastocyst development is presented. Odds ratio (OR) <1 indicate a negative effect of treatment.

The effect of treatment (PFOS, PFHxS) on ordinal variables (stage, grade) were analysed using cumulative link mixed-effect models (clmm model of CRAN-package). Replicate was included as a random factor. Blastocyst stages were condensed to stage 1 (early blastocyst/blastocyst) or 2 (expanded/expanded/hatching/hatched) for analysis.

Cell count and TUNEL-positive cells were log-transformed to estimate normal distribution. Generalized linear mixed models (glmer model of lme4 package) were used to calculate the effect of cell count (blastocyst size) on TUNEL-positive cells as well as the effect of treatment on cell count and TUNEL positive cells. The proportion of TUNEL positive cells were lower in blastocysts with high cell counts compared to those with lower cell counts (see Results section). Therefore, the number of nuclei was accounted for in the model of the effect of treatments (PFOS, PFHxS) on proportion of TUNEL-positive cells.

P-values <0.05 were regarded as significant, and p-values ≥0.05<0.1 were regarded as tendencies.

# Results

## Developmental competence & blastocyst morphology

Of the 855 fertilized oocytes, 341 were cleaved (of which 181 beyond the 2-cell stage) 48 hpf. Further, 158 presumed blastocysts developed. Mean developmental parameters, odds ratios (OR), and p-values categorized by treatment are presented in Table 1. None of the treatments had a significant effect on cleavage rate (p = 0.86 (PFHxS), p = 0.58 (PFOS)), cleavage rate above the 2 cell stage (p = 0.84 (PFHxS), p = 0.48 (PFOS)), or blastocyst rate on day 5 pf (p = 0.07 (PFHxS), p = 0.31 (PFOS)). PFOS did not have an effect on blastocyst rate on day 6 pf (p = 0.17), however PFHxS tended to result in a higher number of blastocysts on day 6 pf compared to the control group (OR: 1.56, p = 0.05). There was no difference in developmental stage (p = 0.21 (PFOS), p = 0.62 (PFHxS)) nor quality grade (p = 0.93 (PFOS), p = 0.56 (PFHxS)) upon PFAS exposure during IVM. See Fig 1.

**Table 1. Developmental competence parameters.**

| | Control | PFHxS, 40 μg/mL | | | PFOS, 0.1 μg/mL | | |
|---|---|---|---|---|---|---|---|
| | Mean (SD) | Mean (SD) | OR (CI) | p-value | Mean (SD) | OR (CI) | p-value |
| Cleaved | 0.38 (0.09) | 0.39 (0.15) | 1.1 (0.78–1.54) | 0.86 | 0.38 (0.17) | 1.03 (0.73–1.45) | 0.58 |
| Cleaved above 2 cell stage[1] | 0.59 (0.27) | 0.50 (0.23) | 0.86 (0.57–1.30) | 0.84 | 0.59 (0.28) | 0.96 (0.64–1.44) | 0.48 |
| Blastocyst day 5 | 0.07 (0.06) | 0.12 (0.07) | 1.69 (0.96–3.04) | 0.07 | 0.09 (0.07) | 1.36 (0.75–2.49) | 0.31 |
| Blastocyst day 6 | 0.15 (0.08) | 0.21 (0.11) | 1.56 (1.01–2.42) | 0.05 | 0.19 (0.09) | 1.37 (0.88–2.14) | 0.17 |

Means and standard deviations (SD) of developmental competence parameters upon exposure to PFASs during 45h IVM, expressed as proportions Odds ratios (OR) and 95% confidence intervals (CI) are given for each developmental parameter.

[1]Calculated from cleaved cell embryos 48 hpf

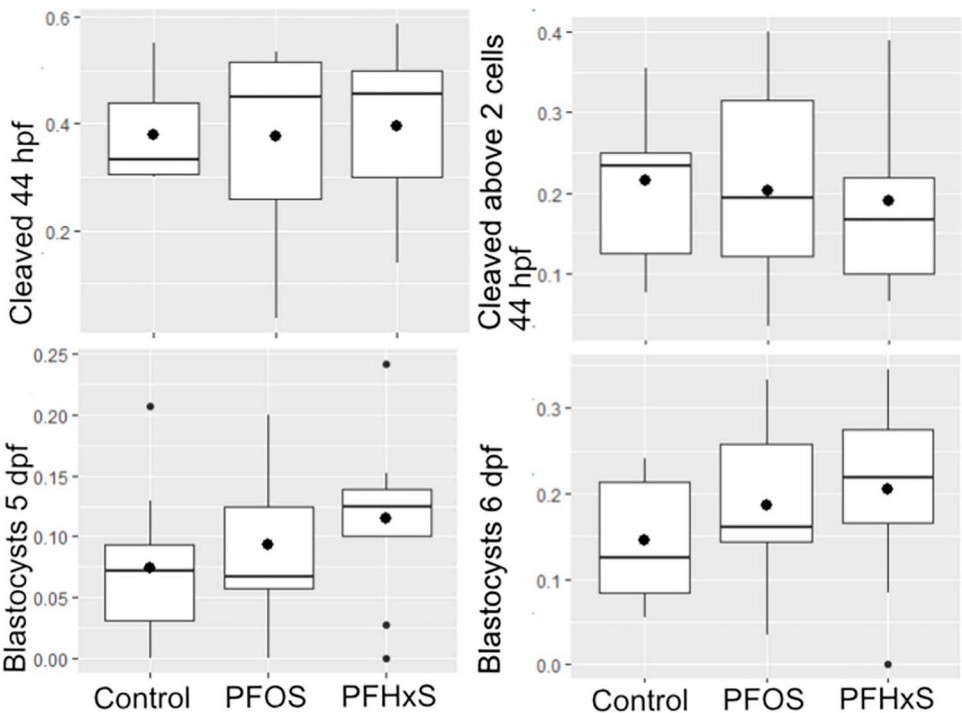

**Fig 1. Developmental competence of embryos.** Box plots over proportion of zygotes cleaved 48 hours post fertilization (hpf, top left), cleaved over the 2-cell stage 48 hpf (top right), proportion of blastocysts 5 days post fertilization (dpf, bottom left) and proportion of blastocysts 6 dpf (bottom right), categorized by treatment (PFOS: 0.1 µg/mL PFHxS: 40 µg/mL).

## Cell count in blastocysts upon exposure to PFAS during IVM

Embryos developed upon exposure to PFASs during IVM with no evident toxicity (see previous section) were stained to detect nuclei (DAPI) and apoptotic cells (TUNEL). The total number of nuclei in blastocysts were as follows: control group mean 35.8 (standard deviation ±17.7); PFHxS group 41.0 (15.2); PFOS group 39.7 (15.6). The increased number of nuclei in embryos treated with PFHxS compared to the control was statistically significant (p = 0.02). For PFOS, the difference was not statistically significant although a tendency was observed (p = 0.07). See Fig 2.

## TUNEL assay

Observation of the positive and negative control blastocysts indicated that the TUNEL assay had labelled apoptotic nuclei (*i.e.* nuclei with exposed 3'-hydroxyl DNA ends) as intended. For the positive controls, the TUNEL staining pattern corresponded to the DAPI staining pattern, and for the negative controls, no stained nuclei were visible in the TUNEL channel. Furthermore, nuclei labelled by the TUNEL assay were typically shrunken/pycnotic in appearance, *i.e.* morphologically implying apoptosis (Fig 3).

## Proportion of apoptotic cells in blastocysts

Embryos containing higher total numbers of cells (*i.e.* larger blastocyst) contained a lower proportion of apoptotic cells (-0.13%, p = 0.002) (Fig 4), which was accounted for in the statistical analyses.

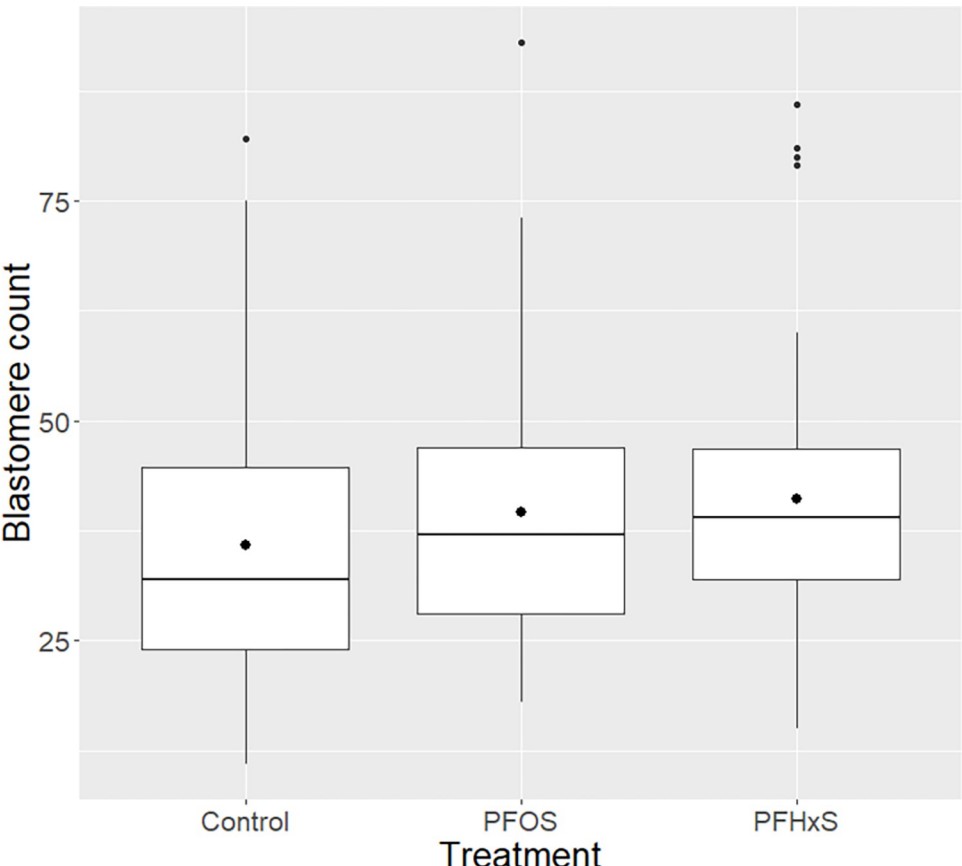

**Fig 2. Blastomere counts.** Box plot over blastomere count in blastocysts, categorized by treatment (PFOS: 0.1 μg/mL, PFHxS: 40 μg/mL). There was a significantly higher number of blastomeres in blastocysts upon exposure to PFHxS during in vitro maturation (p = 0.02), but no significant difference (although a tendency) was seen for PFOS (p = 0.07).

The proportion of apoptotic cells was 10.4% in the control group. Upon PFOS exposure during IVM, there was no significant difference in proportion of TUNEL-positive cells (12.6%, p = 0.19). Upon PFHxS exposure, the resulting blastocysts had a tendency towards a higher proportion of TUNEL-positive cells (13.3%, p = 0.06). See Fig 5 and Table 2.

## Discussion

In this study, porcine oocytes were exposed to PFHxS (40 μg/mL) and PFOS (0.1 μg/mL) during 48h of *in vitro* maturation. This was followed by subsequent embryo development. The concentrations used here were based on previous work within our group on bovine oocytes [29, 34]. The exposure for PFHxS was set higher than for PFOS due to presumed lower toxicity based on our previous findings in the bovine model, where altered early embryonic development upon PFOS and PFHxS exposure was seen at 0.053 μg/mL and >10 μg/mL, respectively [29, 34]. Additionally, earlier research suggests that shorter chained PFASs exert lower toxicity than longer chained equivalents [42].

The concentrations used in this study exceed the levels detected in the average human population. Blood levels of PFASs correlate well with levels in follicular fluid of women [12] and can therefore be used as a proxy for oocyte exposure. Blood levels of PFASs in humans vary depending on compound and geographic region [10]. In one study, median level of PFOS and PFHxS in serum were 5.38 ng/mL and 1.23 ng/mL, respectively [20]. In a second study, levels

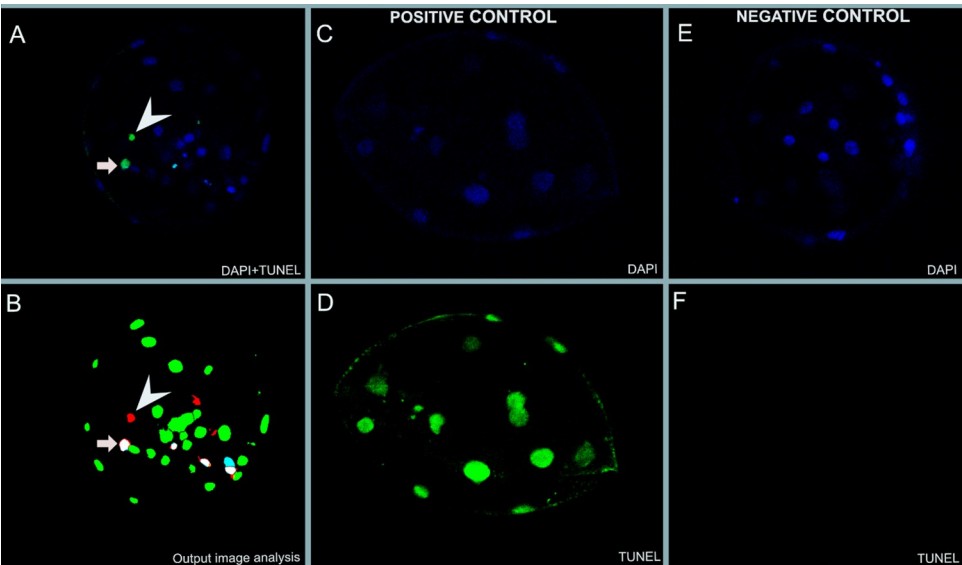

**Fig 3. Stained blastocysts.** Blastocysts stained with DAPI (blue) and TUNEL stain (green). A) shows an overlay section of a blastocyst with DAPI and TUNEL staining, arrow and arrowhead symbolize object detected with TUNEL-assay. B) represents a section of an output from the automated image analysis. Note the cell denoted as apoptotic, where TUNEL and DAPI align (white) compared to the cell where an object is positive for TUNEL, but where positive DAPI is absent (red). In this case, the cell is not annotated as apoptotic. DAPI (C, E) and TUNEL (D, F) of the positive control stained after DNAse treatment and negative control with no TUNEL staining confirm TUNEL assay is labelling as intended.

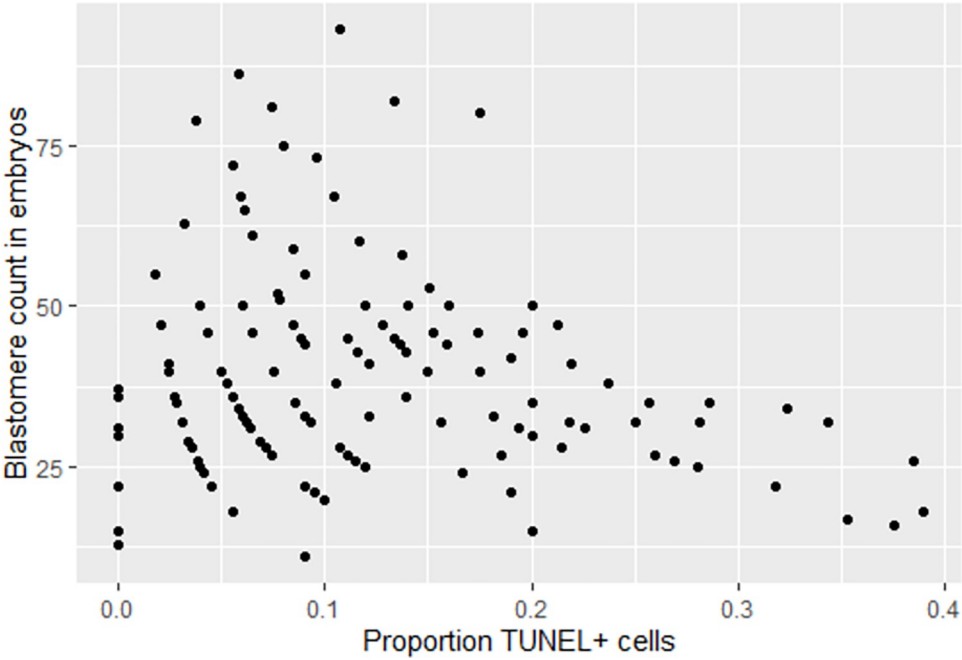

**Fig 4. TUNEL positive cells depending on blastomere count.** Scatterplot of proportion TUNEL positive cells depending on blastomere count in blastocysts. With increasing blastocyst size (increasing blastomere count), the proportion of TUNEL positive cells decrease (p = 0.002).

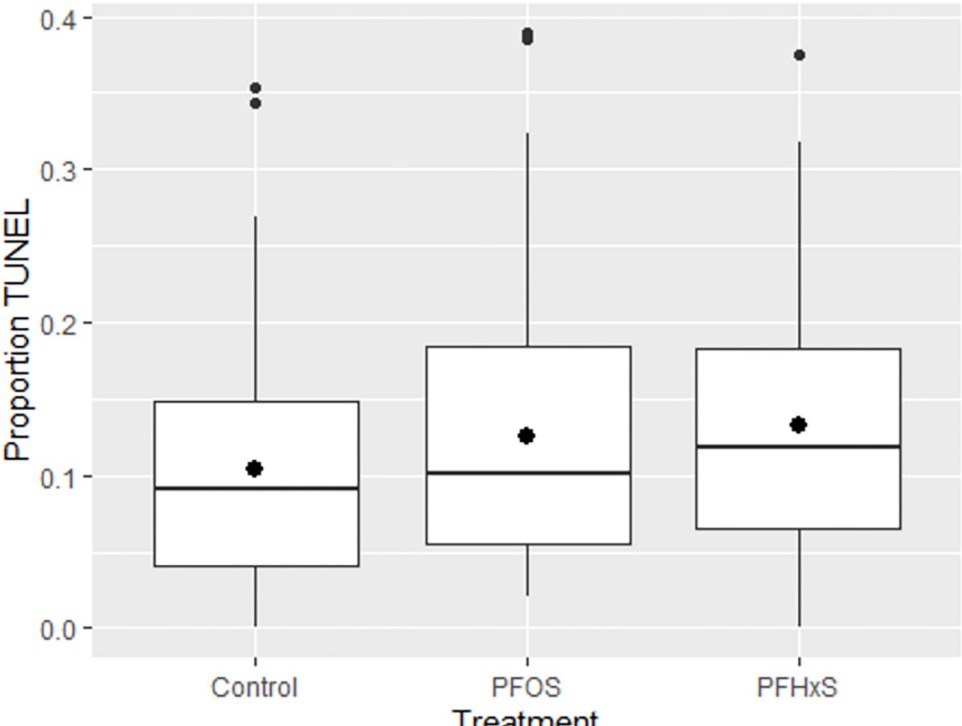

**Fig 5. TUNEL-positive cells in blastocysts.** Box plots over proportion of TUNEL-positive cells for control group, PFOS group (0.1 μg/mL) and PFHxS group (40 μg/mL).

of PFOS and PFHxS in plasma ranged between 1.03–47.8 ng/mL and 0.09–8.46 ng/mL, respectively [43]. However, in heavily contaminated areas, there are reports of substantially higher concentrations in human body fluids [15]. In a cohort of industrial workers in China, some individuals showed serum levels ranging up to 19.8 μg/mL (PFHxS) and 118 μg/mL (PFOS) [16]. Considering the narrow window of exposure used here (oocytes were only exposed during *in vitro* maturation) and possible species differences in sensitivity towards PFASs, it cannot be precluded that findings from studies using higher concentrations may be relevant for humans in general or certainly exposed groups.

Apart from PFHxS appearing to have a positive impact on blastocyst development on day 6, the exposure used did not dramatically affect developmental competence of the oocyte or blastocyst stage/quality grade. Higher concentrations of PFHxS have been showed to inhibit maturation of porcine oocytes [44]. In the work from Martinez-Quezada [44], it was concluded that that 91.68 μM PFHxS inhibited oocyte maturation (40 μg/mL is equivalent to 73 μM).

**Table 2. Proportion of TUNEL-positive cells.**

| Treatment | Mean proportion of apoptotic cells, (SD) | p-value |
|---|---|---|
| Intercept (control) | 0.104 (0.089) | - |
| PFHxS | 0.133 (0.087) | 0.06 |
| PFOS | 0.126 (0.093) | 0.19 |

Means, standard deviations (SD) and p-values for proportion of TUNEL-positive cells in blastocysts 6 days post fertilization, upon exposure to PFHxS and PFOS. Exposure groups are compared with control group.

Cytotoxic effects were seen at 329.1 μM. Interestingly, in the bovine model, exposure to PFHxS during 22h IVM resulted in altered development already at lower concentrations. Developmental toxicity was observed from concentrations $\geq 40$ μg/mL and decreased cell count in blastocysts at concentrations $\geq 20$ μg/mL [29]. Thus, even though the porcine oocytes was exposed for a longer duration (48h of IVM), they seem less sensitive to PFHxS exposure compared to bovine oocytes.

Exposure to PFHxS during oocyte maturation resulted in a tendency towards higher blastocyst count on day 6 post fertilization and significantly increased total cell count in resulting blastocysts. In the bovine model, upon exposure levels below those causing apparent toxicity, a similar response (increased blastocyst stage) was seen. Alterations in genes related to estrogenic pathways could also be observed [29]. Additionally, PFHxS has been showed to be able to exert estrogenic effects [45]. Thus, the increase in blastocyst rate and cell count seen here could potentially be due to an estrogenous effect exerted by PFHxS, while higher concentrations of PFHxS may cause suppressed development. However, mechanisms behind the increased cell count in blastocysts upon PFHxS exposure cannot be determined using the current experimental setup.

We did not see any effects on developmental competence upon exposure to PFOS during *in vitro* maturation. This is in line with previous results seen in murine and porcine oocyte systems using similar concentrations as here [28, 46]. In bovines, slight morphological changes have been observed at lower concentrations (53 ng/mL [34]) despite the fact that bovine oocytes are generally only matured for 24 h, which further implies possible differences in sensitivity between species.

In this study, traditional IVP procedures are combined with objective image analysis. We were able to establish a method to objectively analyse TUNEL positive cells in blastocysts using automated image analysis. The method was deemed satisfactory based on manual validation. Objective methods for image analysis has the potential to make procedures more efficient and to eliminate bias that can be introduced with subjective evaluation. With the use of this method, we were able to show that overall, larger embryos contained lower proportions of apoptotic cells. This could be expected since embryos developing at a faster rate could be assumed to have a greater potential for continued growth.

In contrast to some earlier results seen in other *in vitro* systems [30, 31] no effects on apoptosis attributable to PFOS exposure were seen (as visualized in Figs 1, 2 and 5). This could potentially be explained by lower sensitivity towards PFOS in pigs compared to other species (see discussion regarding species differences above). When it comes to PFHxS, only a tendency towards increased apoptosis rate was seen. Previously, PFHxS has been linked to possible alteration of genes associated with pathways related to apoptosis and oxidative stress [29]. Although there is too little evidence to draw firm conclusions, the effect of PFASs on apoptosis should be studied further.

In this study, a porcine *in vitro* model was used to model early embryo toxicity without the use of experimental animals. Hence, the use of *in vitro* models could contribute to implementation of 3R strategies in research. Ultimately, the developmental competence of an oocyte is not proven until the birth of a healthy offspring, and the *in vitro* setting can only be used to evaluate development until the blastocyst stage. As embryos are only cultured for a limited period of time after fertilization, it is not possible to study effects of PFASs on continued development of the embryo, foetal development, or live offspring when using *in vitro* models. Furthermore, although the pig could be regarded as physiologically similar to humans when compared to e.g. rodents, the possibility of species differences in responses to PFAS exposure cannot be precluded.

## Conclusion

In summary, this study indicates that PFHxS impacts early embryonic development by increasing the total cell number in blastocysts. It is still unclear whether PFHxS impacts apoptosis rates in porcine blastocysts, and although this study adds to the evidence of a possible effect, further research is needed to be able to draw firm conclusions. The concentration of PFOS used here did not impact any of the parameters studied. However, earlier experiments have revealed negative effects on reproduction. Further studies are needed to investigate implications for continued embryonic and foetal development as well as public health.

Finally, the current research provides insightful interpretation of coming trends in assisted reproductive technologies (ARTs) targeted at recognizing the ectopic multi-functional molecular factors (PFOS and PFHxS) that represent a family of PFASs and act as endocrine disruptors, inducers of gameto- and embryotoxicity and promoters/agonists of apoptotic cell death. The use of these factors for assisted reproductive technologies can contribute to attenuation of cytological quality of porcine IVF-derived embryos by augmented incidence of TUNEL-positive (i.e., late-apoptotic) cells in the blastocysts generated under the in vitro culture conditions. The results of these investigations might be extrapolated to studies focused on the IVP of porcine and other mammalian embryos that have been generated by such innovative assisted reproductive technologies as intracytoplasmic sperm injection (ICSI)-mediated IVF [47–50] and somatic cell nuclear transfer (SCNT)-mediated cloning [51–56].

## Supporting information

**S1 File. File containing data used for statistical analysis.**
(XLSX)

## Author Contributions

**Conceptualization:** Ylva Cecilia Björnsdotter Sjunnesson, Ida Hallberg.

**Data curation:** Anna Leclercq, Petter Ranefall, Ylva Cecilia Björnsdotter Sjunnesson.

**Formal analysis:** Anna Leclercq, Petter Ranefall, Ida Hallberg.

**Funding acquisition:** Ylva Cecilia Björnsdotter Sjunnesson, Ida Hallberg.

**Investigation:** Anna Leclercq, Ida Hallberg.

**Methodology:** Petter Ranefall, Ylva Cecilia Björnsdotter Sjunnesson, Ida Hallberg.

**Project administration:** Ylva Cecilia Björnsdotter Sjunnesson, Ida Hallberg.

**Resources:** Ylva Cecilia Björnsdotter Sjunnesson.

**Software:** Petter Ranefall.

**Supervision:** Ylva Cecilia Björnsdotter Sjunnesson, Ida Hallberg.

**Validation:** Anna Leclercq, Petter Ranefall.

**Visualization:** Anna Leclercq, Petter Ranefall.

**Writing – original draft:** Anna Leclercq.

**Writing – review & editing:** Anna Leclercq, Petter Ranefall, Ylva Cecilia Björnsdotter Sjunnesson, Ida Hallberg.

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
