## [Decision Letter · Decision Letter 0]

9 Nov 2022

PONE-D-22-27248Apopotosis in porcine blastocysts upon exposure to perfluoroalkyl substances (PFASs) during oocyte maturation in vitroPLOS ONE

Dear Dr. Leclercq,

Thank you for submitting your manuscript to PLOS ONE. After careful consideration, we feel that it has merit but does not fully meet PLOS ONE’s publication criteria as it currently stands. Therefore, we invite you to submit a revised version of the manuscript that addresses the points raised during the review process.

We look forward to receiving your revised manuscript.

Kind regards,

Hai O. Xu

Academic Editor

PLOS ONE

Journal Requirements:

Reviewers' comments:

Reviewer's Responses to Questions

**Comments to the Author**

1. Is the manuscript technically sound, and do the data support the conclusions?

Reviewer #1: Yes

Reviewer #2: Yes

2. Has the statistical analysis been performed appropriately and rigorously? 

Reviewer #1: No

Reviewer #2: Yes

3. Have the authors made all data underlying the findings in their manuscript fully available?

Reviewer #1: No

Reviewer #2: Yes

4. Is the manuscript presented in an intelligible fashion and written in standard English?

Reviewer #1: Yes

Reviewer #2: Yes

5. Review Comments to the Author

Reviewer #1: In the present study, the authors used porcine oocytes separately cultured in PFOS, PFHxS and control. After the oocytes matured, they were fertilized in vitro. By performing TUNEL assays on the generated blastocysts as well as objective image analysis, the investigators assessed the effects of PFOS and PFHxS on cell counts and the percentage of apoptotic cells in blastocysts. The results showed that PFOS did not affect any parameters at the concentrations used for the experiments in culture. PFHxS affected early embryo development by increasing the total cell count of blastocysts.

This reviewer is satisfied with the significance of this study and the care in which the study was performed. However, there are several major issues to be revised.

1. During oocyte maturation,you need to list detailedly how PFOS,PFHXS and control are Compared.

2. You need to add the toxic mechanism.

3. In the discussion, you need to give the source determined experimentally using the concentration of PFOS and PFHXS. For example, you can insert references or exact evidence. In this way, the use of experimental agents is more representative and scientific.

4.When it comes to the concentration range of PFOS and PFHXS related to workers in Chinese industry, you need to clearly indicate the source of the data or references.

5.For the results of PFOS mentioned in the conclusion section, you need to add clear graphs or references. In this way,it can be visually demonstrated that the concentration of PFOS used in this experiment did not affect any parameters.

6.From the conclusion, there is no innovative breakthrough direction for the research topic of the effect of perfluoroalkyl substances on early embryonic development and blastocyst apoptosis. However, you can focus on the relationship between automated image analysis and the innovative combination of experimentation.

Reviewer #2: 4th November, 2022

Review of the Manuscript No. PONE-D-22-27248, by A. Leclercq et al., entitled: “Apopotosis in porcine blastocysts upon exposure to perfluoroalkyl substances (PFASs) during oocyte maturation in vitro” that is intended to be published as the Research Article in PLoS One

(separate Microsoft Word file as Reviewer Attachment for Manuscript No. PONE-D-22-27248 PLoS One 4th November 2022 that includes Comments to the Authors is also uploaded)

Taking into consideration research highlight, contribution of the Authors to the progress in the research area, thorough manner of data presentation, relatively well writing in English, abundance of Materials and Methods, Results, Tables and Figures (diligent tabular and graphic/photographic documentation), the quality of this paper deserves praise and merits my support. The Authors have received the high scores from me for the originality, importance of the work and the scientific value of their paper. In my opinion, the current paper provides insightful interpretation of topical and coming trends in assisted reproductive technologies (ARTs) targeted at recognizing the ectopic multi-functional molecular factors (perfluorooctane sulfonic acid; PFOS and perfluorohexane sulfonic acid; PFHxS) that represent the group of perfluoroalkyl substances (PFASs) and act as endocrine disruptors, inducers of gameto- and embryotoxicity and promoters/agonists of apoptotic cell death. The use of these factors for ARTs can contribute to attenuation of cytological quality of porcine IVF-derived embryos by augmented incidence of TUNEL-positive (i.e., late-apoptotic) cells in the blastocysts generated under the in vitro culture conditions. For all those reasons, I recommend the Editorial Board to allow for publication of this interesting paper in PLoS One, after the minor revision of the manuscript will have been completed by the Authors and provided that the Authors are ready to consider all the Reviewer comments indicated below:

1) The current title of the manuscript:

“Apopotosis in porcine blastocysts upon exposure to perfluoroalkyl substances (PFASs) during oocyte maturation in vitro”

should be re-edited to one of the following versions:

a) “Occurrence of late-apoptotic symptoms in porcine ex vivo-fertilized embryos upon exposure of oocytes to perfluoroalkyl substances (PFASs) under the conditions of in vitro meiotic maturation”

b) “Exposure of porcine oocytes to perfluoroalkyl substances (PFASs) during their in vitro meiotic maturation triggers late-apoptotic events in the ex vivo-fertilized embryos”

2) In my opinion, there is a lack of finalizing paragraph (including future directions and goals) within the Conclusion section. Therefore, missing details at the end of Conclusion section and missing research article citations and related References are required to be added according to the Reviewer comments shown below:

Considering the aforementioned fact, the following sentences have to be added between the lines 366 and 367 of the Conclusion section (on the page 16) as indicated below:

Finally, the current research provides insightful interpretation of coming trends in assisted reproductive technologies (ARTs) targeted at recognizing the ectopic multi-functional molecular factors (PFOS and PFHxS) that represent a family of PFASs and act as endocrine disruptors, inducers of gameto- and embryotoxicity and promoters/agonists of apoptotic cell death. The use of these factors for ARTs can contribute to attenuation of cytological quality of porcine IVF-derived embryos by augmented incidence of TUNEL-positive (i.e., late-apoptotic) cells in the blastocysts generated under the in vitro culture conditions. The results of these investigations might be extrapolated to studies focused on the in vitro production (IVP) of porcine and other mammalian embryos that have been generated by such innovative assisted reproductive technologies (ARTs) as intracytoplasmic sperm injection (ICSI)-mediated IVF (46–49) and somatic cell nuclear transfer (SCNT)-mediated cloning (50–55).

3) The following 10 References have to be added and cited in the text of manuscript (according to the re-editions required by Reviewer in the above-listed comment 1):

46. Okada T, McIlfatrick S, Hin N, Aryamanesh N, Breen J, St John JC. Mitochondrial supplementation of Sus scrofa metaphase II oocytes alters DNA methylation and gene expression profiles of blastocysts. Epigenetics Chromatin. 2022;15(1):12. doi: 10.1186/s13072-022-00442-x

47. Gorczyca G, Wartalski K, Romek M, Samiec M, Duda M. 2022. The Molecular Quality and Mitochondrial Activity of Porcine Cumulus-Oocyte Complexes Are Affected by Their Exposure to Three Endocrine-Active Compounds under 3D In Vitro Maturation Conditions. Int J Mol Sci. 2022;23(9): 4572. doi: 10.3390/ijms23094572

48. Tsampras N., Kollmann M, Craciunas L. Recombinant versus bovine hyaluronidase for oocyte denudation before intracytoplasmic sperm injection: a systematic review and meta-analysis. J Obstet Gynaecol. 2022;42(2): 301–5. doi: 10.1080/01443615.2021.1893670

49. Hernández-Pichardo JE, Ducolomb Y, Romo S, Kjelland ME, Fierro R, Casillas F, Betancourt M. Pronuclear formation by ICSI using chemically activated ovine oocytes and zona pellucida bound sperm. J Anim Sci Biotechnol. 2016;7:65. doi: 10.1186/s40104-016-0124-6

50. Wiater J, Samiec M, Skrzyszowska M, Lipiński, D. Trichostatin A-Assisted Epigenomic Modulation Affects the Expression Profiles of Not Only Recombinant Human α1,2-Fucosyltransferase and α-Galactosidase A Enzymes But Also Galα1→3Gal Epitopes in Porcine Bi-Transgenic Adult Cutaneous Fibroblast Cells. Int J Mol Sci. 2021;22(3): 1386. doi: 10.3390/ijms22031386

51. Skrzyszowska M, Smorąg Z, Słomski R, Kątska-Ksiażkiewicz L, Kalak R, Michalak E, Wielgus K, Lehmann J, Lipiński D, Szalata M, Pławski A, Samiec M, Jura J, Gajda B, Ryńska B, Pieńkowski M. Generation of transgenic rabbits by the novel technique of chimeric somatic cell cloning. Biol Reprod. 2006;74(6): 1114–20. doi: 10.1095/biolreprod.104.039370

52. Assareh N, Shahemabadi M, Varnosfaderani SR, Jafarpour F, Hajian M, Nasr-Esfahani MH. Sequential IVM by CNP preincubation and cooperating of PGE2 with AREG enhances developmental competence of SCNT reconstructs in goat. Sci Rep. 2022;12(1): 4243. doi: 10.1038/s41598-022-08238-5

53. Samiec M, Skrzyszowska M. Extranuclear Inheritance of Mitochondrial Genome and Epigenetic Reprogrammability of Chromosomal Telomeres in Somatic Cell Cloning of Mammals. Int J Mol Sci. 2021;22(6): 3099. doi: 10.3390/ijms22063099

54. Petersen B, Kurtz S. Generation of Pigs that Produce Single Sex Progeny. Methods Mol Biol. 2022;2495: 275–93. doi: 10.1007/978-1-0716-2301-5_15

55. Opiela J, Samiec M, Romanek J. In vitro development and cytological quality of inter-species (porcine→bovine) cloned embryos are affected by trichostatin A-dependent epigenomic modulation of adult mesenchymal stem cells. Theriogenology. 2017;97: 27–33. doi: 10.1016/j.theriogenology.2017.04.022

4) Finally, the manuscript requires to be slightly revised/corrected for English-language grammar, spelling and style.

General Comment of the Reviewer:

Before the manuscript will have been accepted for publication in PLoS One, it requires the minor revision (according to all the remarks and suggestions of the Reviewer) and confirmation for the correctness of changes that will have been made by the Authors in the re-edited and resubmitted version of their paper.

6. PLOS authors have the option to publish the peer review history of their article (what does this mean?). If published, this will include your full peer review and any attached files.

Reviewer #1: No

Reviewer #2: No

---

## [Author Response · Author response to Decision Letter 0]

7 Dec 2022

Dear reviewers, 

Thank you for your thorough review on our manuscript “Apotosis in porcine blastocysts upon exposure to perfluoroalkyl substances (PFASs) during oocyte maturation in vitro” submitted for potential publication in PlosONE. Please find responses to both reviewer comments below, along with the revised manuscript (clean and tracked changes). 

We were pleased to read the positive feedback about the manuscript and innovative experimental design. We believe that the comments from the reviewers have improved the quality of the manuscript and we thank the reviewers for their effort. 

We hope to hear from you in the near future. 

Best regards, 

Anna Leclercq, on behalf of all authors. 

 

Review comments: 

1. Is the manuscript technically sound, and do the data support the conclusions?

Reviewer #1: Yes

Reviewer #2: Yes

2. Has the statistical analysis been performed appropriately and rigorously? 

Reviewer #1: No

Reviewer #2: Yes

3. Have the authors made all data underlying the findings in their manuscript fully available?

Reviewer #1: No

Reviewer #2: Yes

4. Is the manuscript presented in an intelligible fashion and written in standard English?

Reviewer #1: Yes

Reviewer #2: Yes

5. Review Comments to the Author

1. During oocyte maturation, you need to list detailedly how PFOS,PFHXS and control are Compared.

During in vitro maturation, all oocytes were treated equally in all aspects except for exposure to PFOS and PFHxS. We have added some clarifications in the methods section (line 143-147). All oocytes were cultured to follow early embryo development in vitro, which did not allow further invasive morphological assessment of oocytes after in vitro maturation. However, the oocytes were inspected visually after oocyte maturation, and a subjective assessment of the grade of expansion was assigned to each group. No differences between groups could be seen in any of the replicates. However, because of the subjectivity of this assessment, these data are not presented in the manuscript.

2. You need to add the toxic mechanism.

The current experiment report changes in the phenotype of early embryos upon oocyte exposure to PFAS during oocyte maturation in vitro. Although the toxic mechanisms behind the altered phenotypic changes observed cannot be fully explained using the current experimental setup, we have clarified the discussion where we use previous data on gene expression and DNA methylation to be able to hypothesize on mechanisms of toxicity in the experimental setup used. 

Changes are made in the discussion lines 346-348 and 368-371.

3. In the discussion, you need to give the source determined experimentally using the concentration of PFOS and PFHXS. For example, you can insert references or exact evidence. In this way, the use of experimental agents is more representative and scientific.

Regarding the discussion about the concentrations used in our study, one reference (34) was wrongfully not included. This has now been corrected (lines 126, 131, 316, 319, 508). References are now also indicated more frequently in the discussion (lines 316-319) to make the reasoning more clear

PFOS and PFHxS respectively were analysed to quantify the levels in the stock-solutions used in the experiment, this data has been provided in the M&M section, line 130-131.

4. When it comes to the concentration range of PFOS and PFHXS related to workers in Chinese industry, you need to clearly indicate the source of the data or references.

Thank you for pointing this out. One reference was wrongfully not included in the text. This has now been corrected (line 329).

5. For the results of PFOS mentioned in the conclusion section, you need to add clear graphs or references. In this way, it can be visually demonstrated that the concentration of PFOS used in this experiment did not affect any parameters.

The results upon exposure to PFOS are visualised in figures 1, 2, and 5. As this was not completely clear, a clarifications have been made in the figure legend of Figure 1. The discussion now also includes references to these figures to enhance readability (line 368). 

6. From the conclusion, there is no innovative breakthrough direction for the research topic of the effect of perfluoroalkyl substances on early embryonic development and blastocyst apoptosis. However, you can focus on the relationship between automated image analysis and the innovative combination of experimentation.

We agree that it is important to emphasize the innovative combinations of methods used here, and have added some more lines about this in the discussion (lines 358-364).

1) The current title of the manuscript:

“Apopotosis in porcine blastocysts upon exposure to perfluoroalkyl substances (PFASs) during oocyte maturation in vitro”

should be re-edited to one of the following versions:

a) “Occurrence of late-apoptotic symptoms in porcine ex vivo-fertilized embryos upon exposure of oocytes to perfluoroalkyl substances (PFASs) under the conditions of in vitro meiotic maturation”

b) “Exposure of porcine oocytes to perfluoroalkyl substances (PFASs) during their in vitro meiotic maturation triggers late-apoptotic events in the ex vivo-fertilized embryos”

Thank you, we have carefully considered your suggestions and believe that your first option is an improvement compared to the previous title. However, we changed the wording and shortened the title slightly

2) In my opinion, there is a lack of finalizing paragraph (including future directions and goals) within the Conclusion section. Therefore, missing details at the end of Conclusion section and missing research article citations and related References are required to be added according to the Reviewer comments shown below:

Considering the aforementioned fact, the following sentences have to be added between the lines 366 and 367 of the Conclusion section (on the page 16) as indicated below:

Finally, the current research provides insightful interpretation of coming trends in assisted reproductive technologies (ARTs) targeted at recognizing the ectopic multi-functional molecular factors (PFOS and PFHxS) that represent a family of PFASs and act as endocrine disruptors, inducers of gameto- and embryotoxicity and promoters/agonists of apoptotic cell death. The use of these factors for ARTs can contribute to attenuation of cytological quality of porcine IVF-derived embryos by augmented incidence of TUNEL-positive (i.e., late-apoptotic) cells in the blastocysts generated under the in vitro culture conditions. The results of these investigations might be extrapolated to studies focused on the in vitro production (IVP) of porcine and other mammalian embryos that have been generated by such innovative assisted reproductive technologies (ARTs) as intracytoplasmic sperm injection (ICSI)-mediated IVF (46–49) and somatic cell nuclear transfer (SCNT)-mediated cloning (50–55).

Thank you for the suggestion. We agree that the conclusion lacked a finishing paragraph and have included it. The references indicated below have also been included. 

3) The following 10 References have to be added and cited in the text of manuscript (according to the re-editions required by Reviewer in the above-listed comment 1):

46. Okada T, McIlfatrick S, Hin N, Aryamanesh N, Breen J, St John JC. Mitochondrial supplementation of Sus scrofa metaphase II oocytes alters DNA methylation and gene expression profiles of blastocysts. Epigenetics Chromatin. 2022;15(1):12. doi: 10.1186/s13072-022-00442-x

47. Gorczyca G, Wartalski K, Romek M, Samiec M, Duda M. 2022. The Molecular Quality and Mitochondrial Activity of Porcine Cumulus-Oocyte Complexes Are Affected by Their Exposure to Three Endocrine-Active Compounds under 3D In Vitro Maturation Conditions. Int J Mol Sci. 2022;23(9): 4572. doi: 10.3390/ijms23094572

48. Tsampras N., Kollmann M, Craciunas L. Recombinant versus bovine hyaluronidase for oocyte denudation before intracytoplasmic sperm injection: a systematic review and meta-analysis. J Obstet Gynaecol. 2022;42(2): 301–5. doi: 10.1080/01443615.2021.1893670

49. Hernández-Pichardo JE, Ducolomb Y, Romo S, Kjelland ME, Fierro R, Casillas F, Betancourt M. Pronuclear formation by ICSI using chemically activated ovine oocytes and zona pellucida bound sperm. J Anim Sci Biotechnol. 2016;7:65. doi: 10.1186/s40104-016-0124-6

50. Wiater J, Samiec M, Skrzyszowska M, Lipiński, D. Trichostatin A-Assisted Epigenomic Modulation Affects the Expression Profiles of Not Only Recombinant Human α1,2-Fucosyltransferase and α-Galactosidase A Enzymes But Also Galα1→3Gal Epitopes in Porcine Bi-Transgenic Adult Cutaneous Fibroblast Cells. Int J Mol Sci. 2021;22(3): 1386. doi: 10.3390/ijms22031386

51. Skrzyszowska M, Smorąg Z, Słomski R, Kątska-Ksiażkiewicz L, Kalak R, Michalak E, Wielgus K, Lehmann J, Lipiński D, Szalata M, Pławski A, Samiec M, Jura J, Gajda B, Ryńska B, Pieńkowski M. Generation of transgenic rabbits by the novel technique of chimeric somatic cell cloning. Biol Reprod. 2006;74(6): 1114–20. doi: 10.1095/biolreprod.104.039370

52. Assareh N, Shahemabadi M, Varnosfaderani SR, Jafarpour F, Hajian M, Nasr-Esfahani MH. Sequential IVM by CNP preincubation and cooperating of PGE2 with AREG enhances developmental competence of SCNT reconstructs in goat. Sci Rep. 2022;12(1): 4243. doi: 10.1038/s41598-022-08238-5

53. Samiec M, Skrzyszowska M. Extranuclear Inheritance of Mitochondrial Genome and Epigenetic Reprogrammability of Chromosomal Telomeres in Somatic Cell Cloning of Mammals. Int J Mol Sci. 2021;22(6): 3099. doi: 10.3390/ijms22063099

54. Petersen B, Kurtz S. Generation of Pigs that Produce Single Sex Progeny. Methods Mol Biol. 2022;2495: 275–93. doi: 10.1007/978-1-0716-2301-5_15

55. Opiela J, Samiec M, Romanek J. In vitro development and cytological quality of inter-species (porcine→bovine) cloned embryos are affected by trichostatin A-dependent epigenomic modulation of adult mesenchymal stem cells. Theriogenology. 2017;97: 27–33. doi: 10.1016/j.theriogenology.2017.04.022

4) Finally, the manuscript requires to be slightly revised/corrected for English-language grammar, spelling and style.

We have revised the manuscript and changes are made as seen by tracked changes in the attached manuscript. 

We have also corrected the formatting according to PLOS guidelines.

---

## [Editor Report · Decision Letter 1]

12 Dec 2022

Occurrence of late-apoptotic symptoms in porcine preimplantation embryos upon exposure of oocytes to perfluoroalkyl substances (PFASs) under in vitro meiotic maturation

PONE-D-22-27248R1

Dear Dr. Leclercq,

We’re pleased to inform you that your manuscript has been judged scientifically suitable for publication and will be formally accepted for publication once it meets all outstanding technical requirements.

Kind regards,

Hai O. Xu

Academic Editor

PLOS ONE
---

## [Editor Report · Acceptance letter]

16 Dec 2022

PONE-D-22-27248R1 

Occurrence of late-apoptotic symptoms in porcine preimplantation embryos upon exposure of oocytes to perfluoroalkyl substances (PFASs) under *in vitro* meiotic maturation 

Dear Dr. Leclercq:

I'm pleased to inform you that your manuscript has been deemed suitable for publication in PLOS ONE. Congratulations! Your manuscript is now with our production department. 

Kind regards, 

on behalf of

Dr. Hai O. Xu 

Academic Editor

PLOS ONE